# An In Silico Bioremediation Study to Identify Essential Residues of Metallothionein Enhancing the Bioaccumulation of Heavy Metals in *Pseudomonas aeruginosa*

**DOI:** 10.3390/microorganisms11092262

**Published:** 2023-09-09

**Authors:** Munazzah Tasleem, Wesam M. Hussein, Abdel-Aziz A. A. El-Sayed, Abdulwahed Alrehaily

**Affiliations:** 1School of Electronic Science and Engineering, University of Electronic Science and Technology of China, Chengdu 610054, China; munazzah.t@gmail.com; 2Chemistry Department, Faculty of Science, Islamic University of Madinah, Madinah 42351, Saudi Arabia; wesammeckawy@yahoo.com; 3Biology Department, Faculty of Science, Islamic University of Madinah, Madinah 42351, Saudi Arabia; abdelaziz.a.elsayed@gmail.com

**Keywords:** in silico bioremediation, molecular docking, orthologs, cadmium, lead, *Pseudomonas aeruginosa*

## Abstract

Microorganisms are ubiquitously present in the environment and exert significant influence on numerous natural phenomena. The soil and groundwater systems, precipitation, and effluent outfalls from factories, refineries, and waste treatment facilities are all sources of heavy metal contamination. For example, Madinah, Saudi Arabia, has alarmingly high levels of lead and cadmium. The non-essential minerals cadmium (Cd) and lead (Pb) have been linked to damage to vital organs. Bioremediation is an essential component in the process of cleaning up polluted soil and water where biological agents such as bacteria are used to remove the contaminants. It is demonstrated that *Pseudomonas aeruginosa* (*P. aeruginosa)* isolated from activated sludge was able to remove Cd and Pb from water. The protein sequence of metallothionein from *P. aeruginosa* was retrieved to explore it for physicoparameters, orthologs, domain, family, motifs, and conserved residues. The homology structure was generated, and models were validated. Docking of the best model with the heavy metals was carried out to inspect the intramolecular interactions. The target protein was found to belong to the “metallothionein_pro” family, containing six motifs, and showed a close orthologous relationship with other heavy metal-resistant bacteria. The best model was generated by Phyre2. In this study, three key residues of metallothionein were identified that participate in heavy metal (Pb and Cd) binding, viz., Ala33, Ser34, and Glu59. In addition, the study provides an essential basis to explore protein engineering for the optimum use of metallothionein protein to reduce/remove heavy metals from the environment.

## 1. Introduction

In nature, the presence of microorganisms like bacteria, fungi, archaea, viruses, and microalgae guarantees the retrieval of ecological balance and plays a leading role in the removal of contaminants that hinder biological life. Microorganisms exhibit notable adaptability and fulfill essential functions in diverse ecological processes, encompassing nutrient cycling, primary production, and the breakdown of pollutants [1]. One domain that exhibits considerable promise is the realm of bioremediation, particularly in relation to the elimination of heavy metal pollutants from the natural surroundings. The issue of heavy metal toxicity is a matter of great importance owing to its adverse impacts on human health and the environment as a whole [2]. The presence of a potentially harmful or toxic contaminant in the air, water, or sediment that has an adverse effect on the surrounding ecosystem is known as environmental pollution [3]. The pollutants that reach the water supply via groundwater systems, precipitation, and wastewater outlets from industries, factories, and treatment centers for waste are all direct and indirect sources of heavy metal contamination [4].

In Madinah, Saudi Arabia, Pb and Cd were found at alarming levels in several studies [5,6]. Previous investigations by Maghraby et al. and Ali have extensively evaluated the groundwater quality in the Madinah region, shedding light on the concentrations of heavy metals present in the area. Notably, Maghraby et al. identified the following concentration ranges for heavy metals in the southern Madinah region: arsenic (As) 1–2 µg/L, cadmium (Cd) 1–4 µg/L, copper (Cu) 2–8 µg/L, iron (Fe) 1–37 µg/L, chromium (Cr) 1–146 µg/L, nickel (Ni) 1–18 µg/L, manganese (Mn) 1–39 µg/L, and zinc (Zn) 22–475 µg/L. Similarly, Ali and El-Maghraby reported the following concentration ranges in their study: iron (Fe) 0.046–0.67 mg/L, lead (Pb) 0.0015–0.027 mg/L, manganese (Mn) 0.011–0.48 mg/L, zinc (Zn) 0.011–0.29 mg/L, nickel (Ni) 0.001–0.18 mg/L, copper (Cu) 0.0023–0.0087 mg/L, chromium (Cr) 0.011–0.11 mg/L, cadmium (Cd) 0.0014–0.083 mg/L, selenium (Se) 0.0001–0.017 mg/L, arsenic (As) 0.0001–0.045 mg/L, vanadium (V) 0.002–0.044 mg/L, mercury (Hg) 0.0001–0.0007 mg/L, aluminum (Al) 0.0001–0.11 mg/L, and cobalt (Co) 0.0001–0.0014 mg/L [7,8]. Cd, a non-essential mineral, is potentially risky for vital body organs. [9]. Pb is a toxic metal that has not been assigned a biological role, rather it is widely recognized as an important environmental concern [10]. Cd, a well-recognized industrial and environmental pollutant, finds its way into Madinah through various vectors. It is prevalent in tobacco smoke, commonly used in the region, as well as in fresh and canned fish, and processed vegetable products. Simultaneously, Pb contamination is attributed to multiple sources, including lead additives in motor fuels, the presence of lead water pipes, lead-containing paints, and widespread smoking practices. Furthermore, canned food and drinks have been identified as potential sources of Pb exposure. Notably, Madinah’s rapid industrialization, including activities such as mining, pouring, casting, processing, and metal use, has led to the dispersion of these heavy metals throughout the environment [11,12,13]. Ingesting lead can affect the cardiovascular system and induce other undesired and potentially dangerous changes in the body. The bio-distribution of calcium in bones was also shown to be disrupted by lead exposure [14]. In light of the significant heavy metal contamination in Madinah’s groundwater, the development of effective and sustainable remediation strategies is of utmost importance. Bioremediation, which harnesses the capabilities of microorganisms to degrade or immobilize heavy metals, has emerged as a promising approach. Among the various microorganisms, *Pseudomonas aeruginosa* has shown particular promise in heavy metal bioremediation [1,15,16]. When harmful chemicals need to be eliminated or mitigated, bioremediation can be used by engineered or naturally occurring bacteria. It has been widely acknowledged as an effective and eco-friendly strategy [17]. Metal ions can be effectively removed by using either living or decomposing microbial biomass in a process called biosorption or bioaccumulation [13,14]. Rod-shaped, gram-negative *P. aeruginosa* can be found in various environments, including water, sewage, and hospitals, and also in desert soil, agricultural land, and forests having metal contamination [4]. According to reports, *P. aeruginosa* derived from active sludge was shown to be highly efficient at removing 94.7% of Cd from a solution [5]. In addition, Cd and Pb can be strongly adsorbed to *P. aeruginosa* dead cell debris in water [6]. The environmental isolates of *Pseudomonas* spp. show lead tolerance, which is likely due to their ability to survive in contaminated settings by forming exopolysaccharides and engaging in biosorption [9].

The presented research explored *P. aeruginosa* metallothionein’s family, domain, motifs, and orthologs. The most effective point to begin is with homology models derived from highly similar structural homologs for understanding binding interactions with Pb and Cd in the absence of *P. aeruginosa* metallothionein structure. The model structures were generated using bio-computational methods to explore their structure and functional features. Different structure validation techniques were used to check the models for accurate representations of protein structure. Homology modeling and docking studies offer structural details particularly because structure determines function. Protein active site residues and binding cavity locations were determined through structural and sequence alignment. Docking experiments were conducted on the most optimal model in order to investigate the essential residues and their interaction patterns that play a role in the binding of Pb and Cd with metallothionein. The work illuminates the structural properties and heavy metal binding mechanism of *P. aeruginosa* metallothionein. These findings offer significant guidance for choosing *P. aeruginosa* for bio-remediation procedures.

## 2. Materials and Methods

### 2.1. Sequence Retrieval and Analysis

The *P. aeruginosa* metallothionein protein sequences in FASTA format were obtained through the National Center for Biotechnology Information’s (NCBI) digital information retrieval system (https://www.ncbi.nlm.nih.gov/, accessed on 15 December 2022) [10,14]. Using Expasy’s ProtParam service (https://web.expasy.org/protparam/, accessed on 15 December 2022), the physiochemical properties of metallothionein protein sequences were measured in extensive detail. Antimicrobial drug development must begin with the identification of the virulent protein, and a key step in this process is the determination of the protein’s virulence factor. Therefore, a non-pathogenic protein with the ability to bind heavy metals, especially Pb and Cd, is required. Using the potent VICMPred (https://webs.iiitd.edu.in/raghava/vicmpred/, accessed on 15 December 2022) [17] algorithm, the virulence of protein sequences was determined. The three-dimensional structure of heavy metal Pb and Cd was retrieved through PubChem (https://pubchem.ncbi.nlm.nih.gov/, accessed on 15 December 2022) [18] with PubChem CID: 5352425, and 23973, respectively.

Various bioinformatics techniques were utilized to precisely assign protein functions. Pfam 35.0 (http://pfam-legacy.xfam.org/, accessed on 16 December 2022) [19] and InterPro (https://www.ebi.ac.uk/interpro/, accessed on 16 December 2022) [20] were used to locate the protein sequence families. The MEME v. 5.5.4 tool, available at https://meme-suite.org/meme/, accessed on 16 December 2022 [21], was employed to identify recurring patterns within the query sequence. This is achieved through the utilization of an expectation-optimizing algorithm within a two-component finite hybrid algorithm. Secondary structure prediction aids in homology modeling. The secondary structures were identified in the target sequence by using PsiPred v4.0 (http://bioinf.cs.ucl.ac.uk/psipred/, accessed on 17 December 2022) [22] and Proteus2 (http://wishart.biology.ualberta.ca/proteus2, accessed on 18 December 2022) [23].

### 2.2. Comparative Genomic Analysis and Orthologs Identification

It is possible to make predictions about genes that are conserved across species by using comparative genomics, which can also be used to discover genes that are specific to a single species. OrthoVenn 3 v3.0 Webserver (https://orthovenn3.bioinfotoolkits.net/, accessed on 20 December 2022) [24], was applied for orthologous analysis to evaluate the orthologous cluster by comparing among the genomes of some bacteria that are particularly useful in heavy metal bioremediation: *Enterobacter cloacae*, *Pseudomonas fluorescens*, *Pseudomonas aeruginosa*, *Klebsiella oxytoca*, *Acinetobacter baumannii*, *Bacillus megaterium*, *and Corynebacterium glutamicum*. The most prevalent heuristic best-match strategy is used by the orthologue clustering tool to find orthologous genes that are conserved between species. OrthoMCL Release 6.17 (https://orthomcl.org/orthomcl/app, accessed on 22 December 2022) is a program that sorts proteins into orthologous groups according to their sequence similarity [25]. The combination of these two tools was employed to identify orthologs of metallothionein from *P. aeruginosa*.

### 2.3. Three-Dimensional Structure Prediction and Validation

To generate the model, several tools, including Phyre2 v.2.0 (http://www.sbg.bio.ic.ac.uk/phyre2/html/page.cgi?id=index, accessed on 15 January 2023) [26], LOMETS v.3 (Local Meta-Threading Server, version 3) (https://zhanggroup.org/LOMETS/, accessed on 15 January 2023) [27], SwissModel (https://swissmodel.expasy.org/, accessed on 15 January 2023) [28], and GALAXY (https://GALAXY.seoklab.org/, accessed on 15 January 2023) [29], were implemented. Phyre2 enhances accuracy, alignment, and detection rate through the utilization of Hidden Markov Model alignment. Consequently, it generates three-dimensional models predicated on the primary amino acid sequence of a target protein [26]. Protein structure prediction has been greatly aided by deep learning methods. Template-based predictions of protein structures and function annotation using LOMETS’ meta-server approach make use of new deep learning threading algorithms [27]. SwissModel is an automated homology modeling webserver [28]. GALAXY protein modeling tools combine bioinformatics and physical chemistry to improve techniques for protein modeling challenges with limited information. GALAXY can solve loop modeling and ligand docking problems that use protein structure and an approximation of the binding pocket’s position [29]. Energy minimization was used on the generated 3D models to reduce the number of steric collisions and strains while keeping the overall structure unchanged. Swiss-PDB Viewer was used to apply the GROMOS96 force field for energy minimization calculations [30]. After 3D model optimization, PSICA (Protein Structural Information Conformity Analysis) (http://qas.wangwb.com/~wwr34/mufoldqa/index.html, accessed on 20 January 2023) [31], ERRAT (https://www.doe-mbi.ucla.edu/errat, accessed on 20 January 2023) [32], ProQ3 v.3.0 (https://proq3.bioinfo.se/, accessed on 24 January 2022) [33], and ProCheck (https://www.ebi.ac.uk/thornton-srv/software/PROCHECK/, accessed on 26 January 2023) [34] were used for model verification.

### 2.4. Visualization of the Structure

The models were visualized using PyMOL v 2.5 (www.pymol.org, accessed on 20 February 2023), a molecular visualization system (The PyMOL Molecular Graphics System, Version 1.8 Schrodinger, LLC, New York, NY, USA). To visualize the atomic interactions Discovery Studio Visualizer V21.1 was used.

## 3. Results

### 3.1. Metallothionein Sequence Analysis

The *P. aeruginosa* metallothionein amino acid sequence (accession number: A0A0A8RR37) was obtained from the UniprotKB database (https://www.uniprot.org/, accessed on 20 January 2023). The sequence is composed of 79 amino acid residues. The InterPro classified this protein in the family-Metallothionein, family 14, prokaryote (Metalthion_fam14_prok), homologous family-Metallothionein domain superfamily (Metalthion_dom_sf).

NCBI-CDD and Pfam identified two domains in metallothionein from *P. aeruginosa*: “Metallothio_Pro domain” that starts from residue 6 to 49, and “HincII-like domain” that starts from 34 to 74 residues. Pfam identified one domain “Metallothio_Pro”, which is a Prokaryotic metallothionein that starts from 5 to 49 residues. The InterPro results revealed that it belongs to the “metallothionein” domain superfamily (IPR017854) and the metallothionein domain superfamily. Disordered region the sequences were found to fall in the C-terminal end from 58 to 79 residues. “Metal-ion binding” was found to be the molecular function linked to the target sequence, and other specific molecular functions include ion binding (for lead, calcium, magnesium, metal chelating activity, alkali metals, and transition metals). The MEME tool was used to discover motifs in metallothionein structure. It is found to be composed of six motifs: NSETCV (2–7), VERDGQHY (20–27), ASGHPQ (33–38), GTTRPQ (51–56), VAEDRQLD (57–64), and LKETFP (67–72).

### 3.2. Secondary Structure Content in the Sequences

The physicochemical parameters of the query sequence provide details about the molecular weight (8215.02), theoretical pI (4.29), instability index (50.65), aliphatic index (37.22), and grand average of hydropathicity (GRAVY) (−0.618). The query sequence’s predicted instability index is larger than 40, which suggests that it might be unstable. By analyzing the sequence using the VICMpred prediction technique, the pathogenic potential of *P. aeruginosa’s* metallothionein was determined to be non-pathogenic. The prediction of the secondary structure is usually reliable and much simpler to solve than the prediction of three-dimensional structure. Predicting the shape of a three-dimensional structure with any degree of precision requires it. In addition, secondary structure prediction has the potential to assess the accuracy of a model built using a (tertiary) structure prediction technique [35]. Proteous2, and PsiPred tool results revealed the absence of signal peptide and transmembrane regions in it. The secondary structure prediction is based on consensus from Psipred2, and PsiPred in Figure 1. The metallothionein sequence contains 66% coils, 23% α-helix, and 11% β-strand. Flexible and conformationally changing proteins, like enzymes, rely heavily on random coils [36].

### 3.3. Comparative Genomic Analysis and Orthologs Identification

Orthologs of metallothionein from *P. aeruginosa* were identified using OrthoMCL Release 6.17 tool, BLASTp, and OrthoVenn3 v3.0. Close orthologs were found in *Gloeobacter violaceus* (WP_011143419.1), *Enterobacter cloacae* (SAJ30964.1), *Acinetobacter baumannii* (SCY79822.1), and *Klebsiella pneumoniae* (SVJ77765.1).

These orthologs were subjected to multiple sequence alignment using ClustalOmega tool. The identified orthologs sequences of metallothionein are closely related; however, metallothionein from *G. violaceus* is comparatively shorter and does not share high similarity with other sequences of interest, as shown in Figure 2.

Through OrthoVenn 3 v3.0 [37], a comparative genome-wide analysis of orthologous genes was carried out. An orthologous cluster analysis of seven bacteria (*E. cloacae*, *P. fluorescens*, *P. aeruginosa*, *K. oxytoca*, *A. baumannii*, *B. megaterium*, and *C. glutamicum*) revealed a role in heavy metal bioremediation [10,38]. There were 6863 orthologous gene clusters shared by all seven species, indicating their conservation within bacteria, while *P. aeruginosa* possessed 243 clusters containing 3070 genes. *K. oxytoca* and *E. cloacae* participated in the maximum number of shared gene clusters among all compared species, with 1003 clusters. A total of 851 gene clusters were shared by *P. aeruginosa* and *P. fluorescens*, representing the second gene cluster similarity observed. Proteins from *P. aeruginosa* CAZ10_00455, CAZ10_07020, CAZ10_27830, and CAZ10_35225 interact with proteins from other species, where the highest number of interacting proteins are from *K. oxytoca*. There is only one protein from *E. cloacae*, as shown in Figure 3. Orthologous gene clusters that are shared by all seven bacterial species have been annotated to determine their function in various biological processes—that describe the function of the gene or gene product in living organisms; molecular functions—that describe the biochemical activity (such as binding to specific ligands or structures); and cellular components—that describe the site of activity within the cell [39], as shown in Table 1 and Table 2.

Maximum likelihood [40] was used to establish ancestral states, and CAFE5 [41] was used to analyze variation in the gene family. One possible factor in phenotypic adaptation is the evolution of gene families [42]. According to the dated phylogenetic tree, divergence took place one million years ago. It was determined that *P. fluorescens* and *P. aeruginosa* are sister species that separated from *A. baumannii* 1000 years ago, as shown in Figure 4.

### 3.4. Structure of Metallothionein from P. aeruginosa

Multiple bioinformatics resources, including Uniprot, ExPasy ProtParam, Proteous, VICMPred, OrthoVenn3 v3.0, etc., were utilized to analyze the metallothionein protein. According to the existing literature, *P. aeruginosa* metallothionein homology modeling has not been explored, despite the urgent need to do so in order to gain a more thorough comprehension of metallothionein structure and its impact on metal ion binding. To fully understand the roles of protein, we need to know how it is folded up in three dimensions (3D). Since experimental methods for determining the 3D structure of proteins, such as electron microscopy, X-ray crystallography, and nuclear magnetic resonance, are labor and cost intensive, computational methods for making such predictions are of paramount importance [43]. To improve our understanding of the world of proteins and their properties, computational structural modeling techniques have shown to be a useful addition to experimental structural biology [28].

The quality of the generated model was analyzed using ProQ3, ERRAT, PSICA, and Procheck. Parameters for each model were analyzed, with summaries for all of them shown in Table 3, including the distributions of and on the Ramachandran map produced by residues other than glycine and proline. Figure 5 shows the results of a comparison between the four software tools Phyre2, Swissmodel, LOMETS v.3, and GALAXY. Phyre2 produced the most acceptable models.

The 79-residue homology-modeled structure of metallothionein begins at Met1 and ends at Pro79. The model was observed to contain two β-strands and two α-helices and connected by loops. It is found to contain two domains and six motifs as shown in Figure 6.

Metallothionein has a -CDocker score of 0.0660508 for Pb and a -CDocker Interaction Energy of 0.0660508 for Pb, while Cd has a score of 59.9838 and a -CDocker Interaction Energy of 43.948 for Cd. The distance between Pb and the Ala33 O atom was measured to be 2.15 Å. Figure 7 depicts the electrostatic interactions formed by Cd with the OE2 of Glu59 and the O atom of Ser34, at a distance of 1.64 Å and 2.11 Å, respectively.

Potential bioremediation technologies include biostimulation and other bioremediation strategies [44]. This technique involves supplementing the soil and water supply with nutrients to stimulate the development of microorganisms. Although alteration enhances bioremediation, an overabundance of biomass can block subsurface pores, reducing cleanup efficiency. Remediating bacteria and enzymes hampered by mixed garbage. As a result, the biostimulation of those regions is useless. Genetic and protein-engineering methods are one possible solution to these issues [45]. By using specific promoters, slow-growing bacteria can fully express their preferred genes, preventing biomass accumulation and clogging [46,47].

## 4. Discussion

Sequence analysis has uncovered numerous subgroups, families, and isoforms within the metallothionein superfamily. Polypeptides that exhibit spectroscopic manifestations suggestive of metal thiolate clusters have a unique sequence with a specific organization of cysteines and have a low molecular weight similar to equine renal metallothionein, which are classified as members of the metallothionein superfamily [48]. Metallothioneins that are assumed to be evolutionarily linked and possess specific sequence-specific characteristics that are grouped together as metallothionein families. Prokaryota metallothioneins make up Family 14. The sequence pattern K-C-A-C-x(2)-C-L-C can be used to identify its members. *Cyanobacteria* are included in the members’ taxonomic spectrum. A number of defining features have been identified, including 53–56 AAs, 9 conserved Cys, 1 conserved tyrosine residue, 1 conserved histidine residue, and the presence of other distinctive residues. The structural domain of metallothioneins from both eukaryotes and prokayotes is represented by this superfamily [49]. These proteins have a duplicated metal-bound fold comprised of distinct structural/sequence repeats. Protein sequence motifs are thought to be the defining characteristic of protein families, making it easier to hypothesize about proteins’ potential roles in the body. Because of their connection to catalytic processes, motifs are essential to the performance of enzymes. The identified motifs revealed conserved residues that are participating in the interaction with the heavy metal.

The ProtParam program was used to determine a number of chemical and physical properties, such as the protein’s molecular weight (mg/mol), theoretical pI, amino acid composition, atomic composition, extinction coefficient, estimated half-life, instability index, aliphatic index, and GRAVY (grand average of hydropathicity). The theoretical pI value of the metallothionein protein is determined by adding together the average isotopic masses and linear amino acid pK values, respectively. The pI value of 4.29 indicates the model’s consistency and precision [50]. The aliphatic index is a measure of how much space is occupied by the side chains of aliphatic amino acids (Ala, Val, Ile, and Leu) in a protein. It could be seen as a contributing element to globular proteins’ increased thermostability [51]. A high aliphatic index indicated that the residues along the molecule’s side chains were quite stable and occupied a large volume. Adding up the hydropathy values of each amino acid and then dividing by the total number of amino acid residues yields the GRAVY value for a peptide or protein. Hydrophilic protein structures have negative GRAVY values, and hydrophobic ones have positive ones [52]. The metallothionein protein is hydrophilic, as indicated by its negative GRAVY score (−0.618). In most cases, secondary structure predictions are precise, and their associated problems are much simpler to solve than those associated with three-dimensional structures. It is essential for making accurate projections about components in three dimensions. The accuracy of a model constructed using a (tertiary) structure prediction method can also be evaluated using secondary structure prediction [15,35].

Comparative genomics is one method for predicting the genes shared by multiple species and identifying the genes unique to a particular species. Inferring orthology is the most precise way to characterize similarities and differences within a genome, and the homologous genes (orthologue and paralogue) associated with speciation or duplication are crucial for tracing an organism’s evolutionary pattern [53]. The synteny of orthologous genes between closely related species allows for the identification of a common ancestor during periods of speciation. If orthologous sequences are similar in multiple species, they may have similar biological functions. If they are divergent, they may have different functions in particular species [16,54]. Therefore, determining gene sets that are orthologous and their degree of similarity is crucial in the comparative genomics of two different species for deciphering the evolution of genes and genomes [55]. Orthologs share a similar level of functional specialization. Our understanding of the amino acids that control the specificity of the interactions between protein and ligand relied on this idea. Identification of these residues is a necessary first step in the study of molecular recognition mechanisms and the development of useful proteins and therapies [56]. For this, we combined the maximum likelihood approach [40] with CAFE5 [41] to reconstruct ancestral distributions. For each ancestor node, several amino acids (states) with varying approximations of likelihood at site 1 are displayed. *P. aeruginosa* and *P. fluorescens* are the closest species are close to *A. baumannii*; however, *E. cloacae* and *K. oxytoca* are found to be sister species in the orthologous analysis, which is second the nearest cluster to the *P. aeruginosa*. The protein sequence of metallothionein from close orthologs species identified through OrthoVenn3 is not available; therefore, the orthologous protein sequences were retrieved by applying OrthoMCL and BLASTp that suggests the conservation of the motifs and interacting residues.

Computational three-dimensional structure prediction is faster and more affordable. Model accuracy is affected by goals and prediction methods. PSICA (Protein Structural Information Conformity Analysis) is a fast and accurate web tool for protein model evaluation and accurately evaluates model precision. The accuracy with which a tertiary model represents a primary sequence in a predicted protein is evaluated by comparing it to the structures of known proteins with similar sequences. PSICA makes use of CASP12’s MUfoldQA_S [31]. All the modeled structures have comparably good PSICA scores. The ProQ3 score of the LOMETS model is the highest; however, the Phyre2 has the maximum number of residues in the most favorable region. The G-factor below −0.5 represents an unusual structure [57], revealing the good structure quality of the model generated by Phyre2. The overall protein structure validation analysis revealed the model generated by Phyre2 as the best model for further consideration.

Cadmium and lead are more toxic due to their propensity to bioaccumulate [58]. *P. aeruginosa*, which produces metallothionein, is one of many types of lead- and cadmium-resistant bacteria found in a wide range of environments [59]. It is a biosorbent and has the potential to purify polluted water and soil by binding to and absorbing cadmium and other heavy metals [38]. The metabolic process known as bioaccumulation is active, energy-intensive, and intracellularly bound to metals, which involves metallothionein and metal binding. Low molecular weight, cysteine-rich proteins called “metallothioneins” help toxic metals to be bioaccumulated or sequestered inside of cells. This resistance mechanism is frequently carried by plasmids, which makes it easier for it to spread from one cell to another. Additionally, in reaction to elevated metal exposure, bacteria produce metallothioneins [10,60]. The metallothionein from metallothionein-producing microbes can be used for bioremediation of lead in polluted environments [10]. The in silico analysis revealed the binding of Pb with the conserved residue Ala33. The interactions formed are less than 2.5 Å, indicating close intra-molecular interactions between the Pb and O atoms of Ala33. Other conserved residues Ser34 and Glu59 are participating in close intra-molecular interactions with Cd along with a high −CDOCKER score, indicating good binding of the metals with the metallothionein. Amino acids like Alanine (Ala), Serine (Ser), and Glutamic Acid (Glu) have associations with heavy metal binding due to their distinct chemical properties. Alanine can participate through its backbone amine and carbonyl groups, while Serine’s hydroxyl group can form coordination bonds with metal ions. Glutamic Acid’s carboxylate groups, with their negative charge, strongly interact with positively charged metal ions. These amino acids are integral in the coordination chemistry of heavy metals in biological systems, influencing processes such as metal transport, detoxification, and enzymatic catalysis. The in silico docking highlights the crucial role of Ala33, Ser34, and Glu59 in Pb and Cd binding. Recent investigations have shed light on the intricate nature of *P. aeruginosa’s* interactions with heavy metals and related compounds, emphasizing the significance of acknowledging this complexity. Although our main emphasis lies in exploring the bioremediation capabilities of this bacterium in relation to cadmium and lead, it is important to acknowledge the wider range of metal-related responses exhibited by this organism. Research on the protein PA4063 has revealed its distinctive zinc-binding characteristics, indicating potential functions as a periplasmic zinc chaperone or concentration sensor [61]. Additionally, the examination of solute-binding proteins (SBPs) in *P. aeruginosa* reveals their extensive range of abilities to bind different ligands, such as amino acids, polyamines, and quaternary amines [62]. Furthermore, the examination of the bacterium’s resistance to calprotectin (CP) provides insights into its capacity to flourish in the existence of metal-sequestering proteins, potentially via the mechanism of metallophores [63]. The results of this study highlight the complex and flexible responses of *P. aeruginosa* to various metal challenges. These findings enhance our understanding of the interactions between this bacterium and metals, and their potential significance in bioremediation and environmental adaptation. In light of this, it is clear that *P. aeruginosa* has the capacity to bioaccumulate Pb and Cd to a large extent in order to mitigate their toxicity. For its removal, the bioaccumulation process can be used efficiently. The successful removal of Pb and Cd from contaminated environments utilizing these bacteria can benefit from future studies on heavy metal bioaccumulating microorganisms.

## 5. Conclusions

Numerous lead pollution studies worldwide show the necessity for ecologically friendly heavy metal remediation technology. Lead- and cadmium-resistant bacteria have developed several strategies to withstand Pb and Cd toxicity. Naturally occurring, multi-pollutant-resistant bacteria may be the best option for on-site heavy metal remediation since they can survive additional stresses at polluted places. *P. aeruginosa* is a highly adaptable microorganism with the ability to extract heavy metals from contaminated environments. The comparative genomic analysis reveals 190 protein sequences common among *P. aeruginosa* orthologs. Additionally, *P. fluorescens* and *A. baumannii* are the closest orthologs to *P. aeruginosa*. The *P. aeruginosa* metallothionein was found to contain metal binding domains that assist in binding with heavy metals. The present in silico research establishes the verified three-dimensional structure of *P. aeruginosa* metallothionein. Molecular docking analysis revealed the crucial conserved residues required for binding with Pb and Cd. Ala33 is observed to bind with Pb at a close distance of 2.16 Å, whereas Cd binds to Ser34 and Glu59 at 2.11 and 1.65 Å. In order to create stable metallothionein that can endure and reduce Pb and Cd in the hard climatic circumstances in Madinah, Saudi Arabia, protein engineering can further confirm these interacting key residues.

## Figures and Tables

**Figure 1 microorganisms-11-02262-f001:**
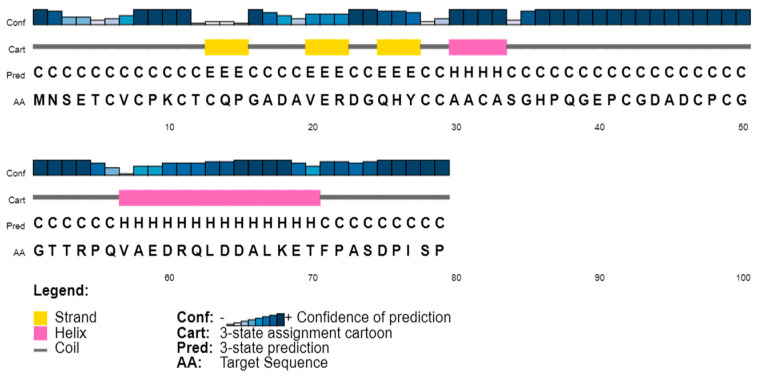
Secondary structure prediction.

**Figure 2 microorganisms-11-02262-f002:**
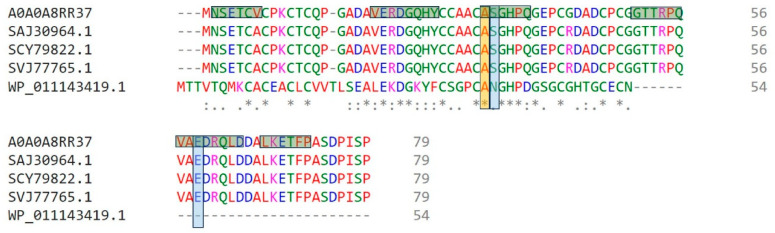
Multiple sequence alignment of metallothionein from orthologous species. The highlighted columns are showing the conserved residues participating in heavy metal interaction. The horizontal green highlights are displaying the motifs. Symbols at the bottom indicate conservation and similarity levels: ‘*’ (Asterisk) for complete conservation, ‘:’ (Colon) for moderate similarity, and ‘.’ (Period) for weak similarity between aligned sequences.

**Figure 3 microorganisms-11-02262-f003:**
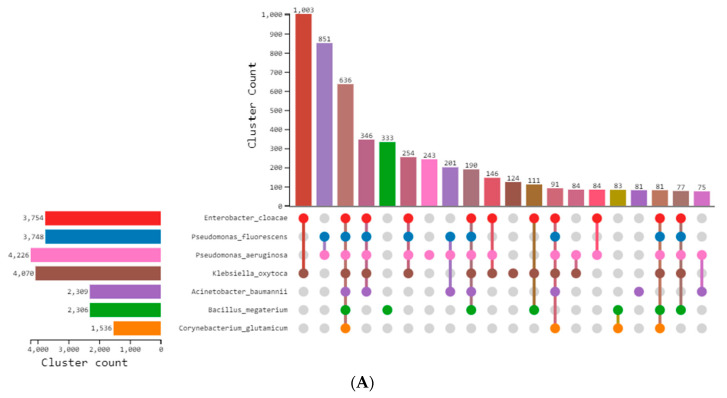
(**A**) The UpSet table lists the number of homologous gene clusters that are both exclusive to each species and shared by all species. The figure legend explains the number of proteins in each species. (**B**) Venn diagram displaying the distribution of common orthologue clusters among species. The symbol “?” represents unclassified proteins not fitting in the orthologous groups. (**C**) Protein interaction network of the orthologous species.

**Figure 4 microorganisms-11-02262-f004:**
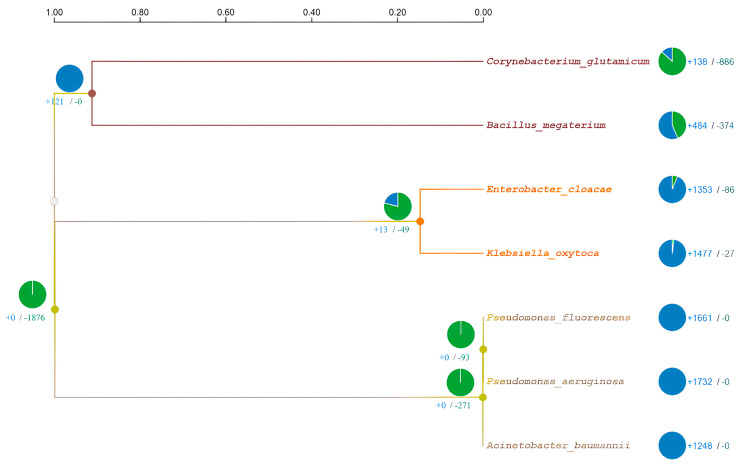
Reconstructed maximum likelihood phylogeny using 37,686 all-protein samples from the seven species, with dated nodes from Time-Tree, and bootstrap index on the left side of nodes. The divergence clock is presented in million years ago at the top. The pie chart represents gene family expansion (blue) and contraction (green).

**Figure 5 microorganisms-11-02262-f005:**
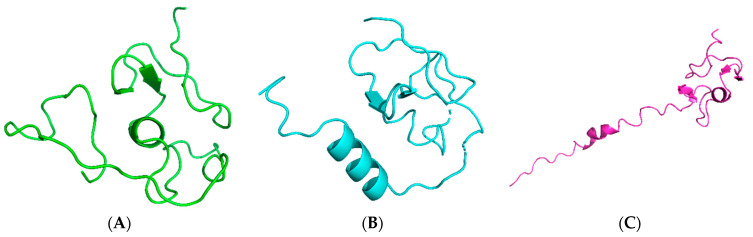
Three-dimensional modeled structures shown in cartoon presentation. (**A**) Model generated by GALAXY webserver, (**B**) model generated by LOMETS, and (**C**) model generated by SwissModel.

**Figure 6 microorganisms-11-02262-f006:**
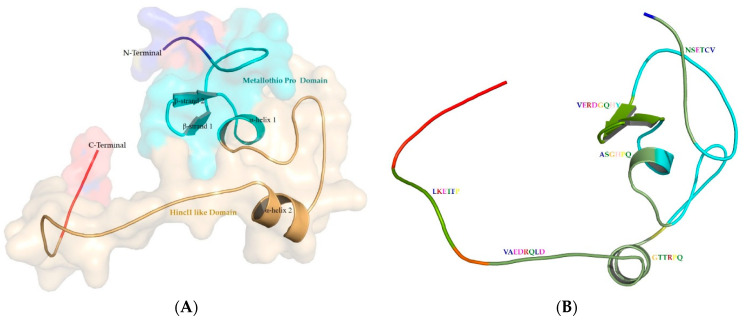
Metallothionein homology model. (**A**) Cartoon representation embedded on the surface view of Metallothionein (*P. aeruginosa*), showing two major domains. (**B**) Identified motifs in metallothionein represented in green color, drawn in PyMol.

**Figure 7 microorganisms-11-02262-f007:**
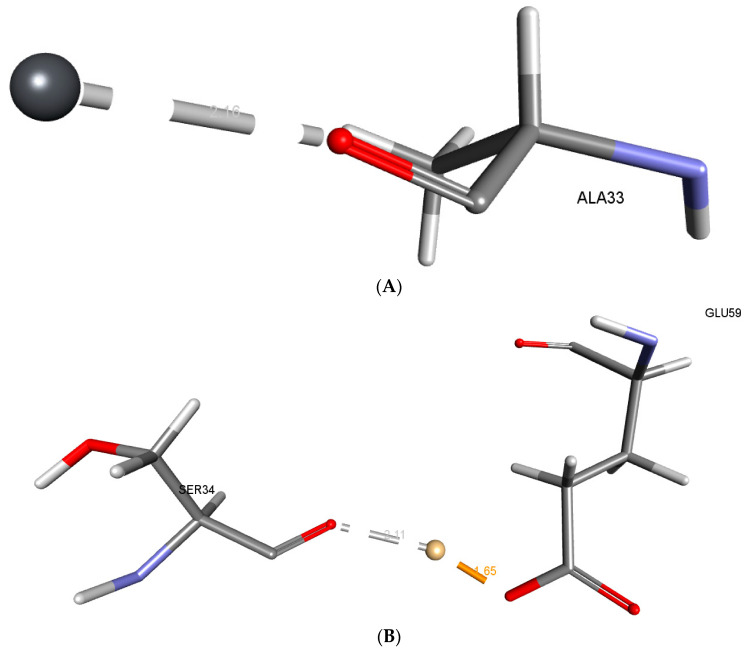
(**A**) Interaction with Lead. (**B**)Interaction with Cd.

**Table 1 microorganisms-11-02262-t001:** Molecular functions associated with the cluster of all seven species.

Slimmed GO	Count of Unique Input Access	Molecular Function
GO:0003674	150	Molecular Function
GO:0016787	138	Hydrolase Activity
GO:0005215	129	Transporter Activity
GO:0003676	113	Nucleic Acid Binding
GO:0016740	109	Transferase Activity
GO:0016491	96	Oxidoreductase Activity
GO:0043167	70	Ion Binding
GO:0008233	55	Peptidase Activity
GO:0000166	41	Nucleotide Binding
GO:0005488	36	Binding

**Table 2 microorganisms-11-02262-t002:** Biological functions associated with the cluster of all seven species.

Slimmed GO	Count of Unique Input Access	Biological Function
GO:0008152	1967	Metabolic Process
GO:0008150	1751	Biological Process
GO:0044237	1724	Cellular Metabolic Process
GO:0006807	1442	Nitrogen Compound Metabolic Process
GO:0044238	1180	Primary Metabolic Process
GO:0009987	946	Cellular Process
GO:0046483	866	Heterocycle Metabolic Process
GO:0006082	783	Organic Acid Metabolic Process
GO:0006725	781	Cellular Aromatic Compound Metabolic Process
GO:0043170	692	Macromolecule Metabolic Process

**Table 3 microorganisms-11-02262-t003:** Metallothionein-generated structure validation.

Modeling Tool	Phyre2	SwissModel	GALAXY	LOMETS
Residues Built	1–79	1–79	1–79	1–79
ProQ3	0.345	0.37	0.401	0.447
ERRAT	47.76	81.63	78.57	44.92
PSICA Server	0.49	0.47	0.53	0.43
Most Favored	85.5	71	83.9	66.1
Additionally Allowed	11.3	25.8	11.3	25.8
Generously allowed	3.2	3.2	1.3	6.5
G-factor	−0.07	−0.36	−0.13	−7.89

## Data Availability

The research reported in the article used no data.

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
