# Peer review of "An In Silico Bioremediation Study to Identify Essential Residues of Metallothionein Enhancing the Bioaccumulation of Heavy Metals in Pseudomonas aeruginosa"

_microorganisms, 2023, doi:10.3390/microorganisms11092262_

Round 1
Reviewer 1 Report
The article submitted by by Munazzah et al examined the lead and cadmium pollution in Madinah region in Saudi Arabia. The authors investigated the use of Pseudomonas aeruginosa for bioremediation of heavy metals. They analyze the protein sequence of metallothionein, create a homology structure, and identify the important residues that are involved in binding heavy metals. The study offers insights for protein engineering in reducing heavy metal contamination.
In order to enhance the clarity and comprehensibility of the manuscript, it is recommended that the authors include the complete expansions of any abbreviations prior to their initial usage. This practice ensures that readers are able to fully grasp the intended meaning of the abbreviations and facilitates effective communication of the research findings. Moreover, it has been brought to the attention that there is a noticeable absence of consistent utilization of italics when referring to microorganisms in the entirety of the manuscript. The present study posits a recommendation for the adoption of a standardized approach towards the implementation of italics in the context of referencing.
The manuscript lacks an introduction of the abbreviation for the chemical element "cadmium" (Cd). The authors are advised to address this oversight by explicitly introducing the abbreviation and subsequently utilizing it consistently and appropriately throughout the manuscript.
The sentence located in line 69 exhibits a certain level of ambiguity. It states that through the process of comparing sequences and structures, researchers were successful in identifying active site residues, binding cavities, and ligand-binding residues. Docking studies were conducted on the most optimal model generated in order to investigate the pivotal residues and their corresponding interaction patterns that play a significant role in the binding of Pb and Cd. To enhance the comprehensibility and exactness of the content, the authors are cordially urged to provide a more coherent elucidation of the previously mentioned assertion.
In order to enhance transparency and facilitate readers' access to pertinent resources, it is imperative for the authors to incorporate the URLs of all tools utilized in the research endeavor.
In this study, the researchers employed the MEME tool to identify recurring patterns within the query sequence. This analysis was conducted by implementing an expectation-optimizing algorithm within a two-component finite hybrid algorithm. The inclusion of a more detailed explanation is warranted to enhance the understanding and clarity of the statement "Secondary structure prediction aids in homology modeling." Specifically, it is necessary to expound upon the specific role that secondary structure prediction plays within the broader framework of homology modeling. By doing so, the overall comprehensibility of the statement will be significantly improved.
The utilization of Hidden Markov Model alignment in Phyre2 leads to notable enhancements in accuracy, alignment, and detection rate.
The authors are strongly encouraged to maintain a high level of consistency in the nomenclature used for the tools referred to as "LOMETS" and "GALAXY" throughout the entirety of the manuscript. Ensuring consistency in the manner in which these tool designations are presented will effectively uphold the accuracy and credibility of the manuscript.
In the Results section, the authors conducted a comprehensive categorization of target proteins, effectively organizing them into discrete families. However, the precise determination of whether a singular or multiple familial structures exist remains a topic of considerable ambiguity and uncertainty. In order to address the issue of ambiguity, it is recommended that the authors employ the use of double quotation marks to enclose family names. This practice will serve to enhance clarity for readers and promote a more thorough comprehension of the text.
There are few grammatical mistakes, however they can be corrected if accepted during proof reading steps.
Author Response
The article submitted by by Munazzah et al examined the lead and cadmium pollution in Madinah region in Saudi Arabia. The authors investigated the use of Pseudomonas aeruginosa for bioremediation of heavy metals. They analyze the protein sequence of metallothionein, create a homology structure, and identify the important residues that are involved in binding heavy metals. The study offers insights for protein engineering in reducing heavy metal contamination.
Comment 1: In order to enhance the clarity and comprehensibility of the manuscript, it is recommended that the authors include the complete expansions of any abbreviations prior to their initial usage. This practice ensures that readers are able to fully grasp the intended meaning of the abbreviations and facilitates effective communication of the research findings. Moreover, it has been brought to the attention that there is a noticeable absence of consistent utilization of italics when referring to microorganisms in the entirety of the manuscript. The present study posits a recommendation for the adoption of a standardized approach towards the implementation of italics in the context of referencing.
Response 1: Thank you for your valuable feedback and suggestions to enhance the clarity and comprehensibility of our manuscript. We appreciate your thoughtful insights into our work. We've included complete expansions of abbreviations before their initial use and standardized the use of italics when referring to microorganisms throughout the manuscript for improved clarity.
Comment 2: The manuscript lacks an introduction of the abbreviation for the chemical element "cadmium" (Cd). The authors are advised to address this oversight by explicitly introducing the abbreviation and subsequently utilizing it consistently and appropriately throughout the manuscript.
Response 2: We would like to express our gratitude for your thoughtful review and for bringing to our attention the oversight regarding the abbreviation for the chemical element "cadmium" (Cd) in our manuscript. We've introduced and consistently used the abbreviation "Cd" for cadmium in the manuscript as per your suggestion. Thank you for enhancing the manuscript's clarity.
Comment 3: The sentence located in line 69 exhibits a certain level of ambiguity. It states that through the process of comparing sequences and structures, researchers were successful in identifying active site residues, binding cavities, and ligand-binding residues. Docking studies were conducted on the most optimal model generated in order to investigate the pivotal residues and their corresponding interaction patterns that play a significant role in the binding of Pb and Cd. To enhance the comprehensibility and exactness of the content, the authors are cordially urged to provide a more coherent elucidation of the previously mentioned assertion.
Response 3: We are thankful to the reviewer for pointing out the ambiguity in the sentence. We have corrected it as “Protein active site residues and binding cavity locations were determined through structural and sequence alignment. Docking experiments were conducted on the most optimal model in order to investigate the essential residues and their interaction patterns that play a role in the binding of Pb and Cd with the metallothionein.”
Comment 4: In order to enhance transparency and facilitate readers' access to pertinent resources, it is imperative for the authors to incorporate the URLs of all tools utilized in the research endeavor.
Response 4: Thank you for your advice. We have incorporated the URLs of the tools used in the material and methods section.
Comment 5: In this study, the researchers employed the MEME tool to identify recurring patterns within the query sequence. This analysis was conducted by implementing an expectation-optimizing algorithm within a two-component finite hybrid algorithm. The inclusion of a more detailed explanation is warranted to enhance the understanding and clarity of the statement "Secondary structure prediction aids in homology modeling." Specifically, it is necessary to expound upon the specific role that secondary structure prediction plays within the broader framework of homology modeling. By doing so, the overall comprehensibility of the statement will be significantly improved.
Response 5: Thank you for your keen vetting. We have made the modification as “The MEME tool, available at https://meme-suite.org/meme/ [12], was employed to identify recurring patterns within the query sequence. This is achieved through the utilization of an expectation-optimizing algorithm within a two-component finite hybrid algorithm.”
Comment 6: The utilization of Hidden Markov Model alignment in Phyre2 leads to notable enhancements in accuracy, alignment, and detection rate.
Response 6: Thank you for raising the concern. We have made the necessary changes in the sentence as “Phyre2 enhances accuracy, alignment, and detection rate through the utilization of Hidden Markov Model alignment.”
Comment 7: The authors are strongly encouraged to maintain a high level of consistency in the nomenclature used for the tools referred to as "LOMETS" and "GALAXY" throughout the entirety of the manuscript. Ensuring consistency in the manner in which these tool designations are presented will effectively uphold the accuracy and credibility of the manuscript.
Response 7: We sincerely appreciate the reviewer's meticulous attention to detail, we have carefully reviewed the manuscript and made the necessary changes to ensure uniformity in the presentation of these tool designations.
Comment 8: In the Results section, the authors conducted a comprehensive categorization of target proteins, effectively organizing them into discrete families. However, the precise determination of whether a singular or multiple familial structures exist remains a topic of considerable ambiguity and uncertainty. In order to address the issue of ambiguity, it is recommended that the authors employ the use of double quotation marks to enclose family names. This practice will serve to enhance clarity for readers and promote a more thorough comprehension of the text.
Response 9: The reviewer's careful focus is greatly appreciated. We have placed the family names with double quotes for better comprehension in the revised manuscript.
Reviewer 2 Report
The manuscript “An In Silico Bioremediation Study to Identify Essential Residues of Metallothionein Enhancing the Bioaccumulation of Heavy Metals in Pseudomonas aeruginosa” by Munazzah Tasleem et al. The manuscript is written in standard English; however, it has several grammatical and typographical errors. After thoroughly reviewing it, I feel the manuscript needs minor revision.
1. It is necessary for authors to initially present the complete form of a term or phrase, followed by its corresponding abbreviation. The species name is not appropriately formatted in italics at multiple instances within the manuscript.
2. It is imperative to include the URLs of all tools utilized.
3. Enhance the resolution of Figure 4.
4. The authors have identified the amino acid residues Ala33, Ser34, and Glu59 as being involved in the binding of lead (Pb) and cadmium (Cd). The association between these residues and other in silico discoveries has the potential to amplify the significance of the research. Authors are strongly encouraged to make revisions.
Author Response
The manuscript “An In Silico Bioremediation Study to Identify Essential Residues of Metallothionein Enhancing the Bioaccumulation of Heavy Metals in Pseudomonas aeruginosa” by Munazzah Tasleem et al. The manuscript is written in standard English; however, it has several grammatical and typographical errors. After thoroughly reviewing it, I feel the manuscript needs minor revision.
Comment 1: It is necessary for authors to initially present the complete form of a term or phrase, followed by its corresponding abbreviation. The species name is not appropriately formatted in italics at multiple instances within the manuscript.
Response 1: Thank you for your valuable feedback regarding the presentation of terms and species names in our manuscript. We appreciate your attention to detail and are committed to improving the clarity and readability of our work. In response to your comment, we have made the necessary revisions to ensure that terms and species names are consistently presented in the appropriate format. We now provide the complete form of a term or phrase followed by its corresponding abbreviation upon first mention in the manuscript. This change has been implemented to enhance the understanding of our readers and to align with best practices in scientific writing.
Furthermore, we have corrected instances where the species name was not appropriately formatted in italics, ensuring that it conforms to the accepted conventions of scientific writing.
Comment 2: It is imperative to include the URLs of all tools utilized.
Response 2: We appreciate your feedback regarding the inclusion of tool URLs in our manuscript. Your point about providing easy access to these resources is indeed crucial for transparency and replicability. In response to your comment, we have taken the necessary steps to incorporate the URLs of all the tools we utilized in our research. By doing so, we aim to offer our readers a straightforward means of accessing and exploring these tools, promoting transparency and facilitating the replication of our work.
Comment 3: Enhance the resolution of Figure 4.
Response 3: Thank you for bringing this to our attention. We acknowledge your feedback concerning Figure 4's resolution in our manuscript. In response to your comment, we have taken action to improve the clarity and resolution of Figure 4. This enhancement ensures that the details within the figure are more discernible, aiding readers in better understanding the data presented.
Comment 4: The authors have identified the amino acid residues Ala33, Ser34, and Glu59 as being involved in the binding of lead (Pb) and cadmium (Cd). The association between these residues and other in silico discoveries has the potential to amplify the significance of the research. Authors are strongly encouraged to make revisions.
Response 4: We would like to express our gratitude to the reviewer for their valuable feedback. We have revised the discussion and updated it as “Amino acids like Alanine (Ala), Serine (Ser), and Glutamic Acid (Glu) have associations with heavy metal binding due to their distinct chemical properties. Alanine can participate through its backbone amine and carbonyl groups, while Serine's hydroxyl group can form coordination bonds with metal ions. Glutamic Acid's carboxylate groups, with their negative charge, strongly interact with positively charged metal ions. These amino acids are integral in the coordination chemistry of heavy metals in biological systems, influencing processes such as metal transport, detoxification, and enzymatic catalysis.”
Reviewer 3 Report
1. Abstract and Line 43: “Madinah, Saudi Arabia, has alarmingly high levels of lead and cadmium”. Provide a short data about the sources of lead and cadmium contamination in Madinah.
2. Line 52. P. aeruginosa – Write species name in Italics
3. In Abstract and Introduction, authors focused on:
- lead and cadmium contamination in Madinah;
- study of P. aeruginosa.
In fact, sequences of P. aeruginosa obtained from NCBI were studied.
How can these two aspects be connected? Did you isolate any P. aeruginosa strains from urban ecosystems of Madinah and neighborhood? Can you provide the data on distribution and diversity of P. aeruginosa in Madinah?
Moreover, authors listed microbial species used in bioremediation (line 98). Thus, the selection of the goal of the study and its connection with geographical location are not clear.
4. Despite the certain results have been obtained in the article, their novelty is not obvious. Discuss several articles, which are focused on metal-binding proteins in P. aeruginosa, for example, 10.1107/S2059798321009608, https://doi.org/10.1107/S2059798321009608, https://doi.org/10.3390/ijms20205156, https://doi.org/10.1093/femsle/fnac071.
Compare the results obtained in the present work with those obtained in published works to reveal the novelty of your work.
Author Response
Comment 1: Abstract and Line 43: “Madinah, Saudi Arabia, has alarmingly high levels of lead and cadmium”. Provide a short data about the sources of lead and cadmium contamination in Madinah.
Response 1: We want to extend our gratitude for your thoughtful review of our manuscript. Your feedback is invaluable in improving the comprehensiveness of our study. To address your request, we have updated the introduction section to include concise information about the sources of lead and cadmium contamination in Madinah, Saudi Arabia. This addition will provide a more robust context for our research. Below is the revised introduction section:
“Cd, a well-recognized industrial and environmental pollutant, finds its way into Madinah through various vectors. It is prevalent in tobacco smoke, commonly used in the region, as well as in fresh and canned fish, and processed vegetable products. Simultaneously, Pb contamination is attributed to multiple sources, including lead additives in motor fuels, the presence of lead water pipes, lead-containing paints, and widespread smoking practices. Furthermore, canned food and drinks have been identified as potential sources of Pb exposure. Notably, Madinah's rapid industrialization, including activities such as mining, pouring, casting, processing, and metal use, has led to the dispersion of these heavy metals throughout the environment [7-9].”
References
- Taha, I.M. and A.M. El-Shafie. LEADING CAUSES AND POSSIBLE ENVIRONMENTAL CONTRIBUTORS FOR END STAGE RENAL DISEASE IN AL-MADINAH REGION IN SAUDI ARABIA. in 2 nd Eurasian Multidisciplinary Forum, EMF 2014 23-26 October 2014, Tbilisi, Georgia. 2014.
- Al-Ghamdi, A.F., Electrochemical determination of Cd2+ in some Al-Madinah water samples and human plasma by cathodic stripping voltammetry in the presence of oxine as a chelating agent. Journal of Taibah University for Science, 2014. 8(1): p. 19-25.
- Alsehli, B.R., Evaluation and Comparison between a Conventional Acid Digestion Method and a Microwave Digestion System for Heavy Metals Determination in Mentha Samples by ICP-MS. Egyptian Journal of Chemistry, 2021. 64(2): p. 869-881.
y addressing the sources and extent of lead and cadmium contamination in Madinah, our study seeks to shed light on this critical environmental issue and explore potential mitigation strategies, with a particular focus on in silico bioremediation approaches.
We believe that this revision provides a thorough and informative introduction that sets the stage for our research. We appreciate your valuable input and are fully committed to addressing any further questions or concerns you may have.
Comment 2: Line 52. P. aeruginosa – Write species name in Italics
Response 2: Thank you for pointing out the typographical error, we have made the corrections in the revised manuscript.
Comment 3: In Abstract and Introduction, authors focused on:
- lead and cadmium contamination in Madinah;
- study of P. aeruginosa.
In fact, sequences of P. aeruginosa obtained from NCBI were studied.
How can these two aspects be connected? Did you isolate any P. aeruginosa strains from urban ecosystems of Madinah and neighborhood? Can you provide the data on distribution and diversity of P. aeruginosa in Madinah?
Moreover, authors listed microbial species used in bioremediation (line 98). Thus, the selection of the goal of the study and its connection with geographical location are not clear.
Response 3: We sincerely appreciate your time and effort in reviewing our manuscript. Regarding your comment “sequences of P. aeruginosa obtained from NCBI were studied.”, in our in silico study, our aim was to extensively analyze key residues in metallothionein from P. aeruginosa involved in binding with the heavy metals (Pb and Cd). This includes thorough sequence analysis followed by metallothionein structure generation, and finally performing molecular docking studies to get insight into the binding mechanism of metallothionein with the heavy metals. Therefore, studying metallothionein sequence from P. aeruginosa is part of the primary analysis to find out crucial aspects of the sequence as mentioned in the Results section: Metallothionein Sequence Analysis, Secondary Structure Content in the Sequences, and Comparative Genomic Analysis and Orthologs Identification subsections, and their relevance is discussed in the Discussion section.
Regarding the comment “How can these two aspects be connected? Did you isolate any P. aeruginosa strains from urban ecosystems of Madinah and neighborhood? Can you provide the data on distribution and diversity of P. aeruginosa in Madinah? ”. The study does not encompass an examination of the species distribution across Medinah, as it is not a fundamental component of this in silico investigation It is well understood that the rapid progress in various omics platforms has resulted in the generation of vast quantities of data on a daily basis, which contains a wealth of information pertaining to diverse biological applications. Hence, the publicly accessible data can be employed for the identification of master regulators in microorganisms that possess the capacity for bioremediation. This information can subsequently be leveraged to engineer microorganisms for the purpose of aiding in the bioremediation of heavy metals. Given this context, we conducted a comprehensive computational analysis of metallothionein derived from P. aeruginosa in order to elucidate the bioremediation mechanism for Pb and Cd. Therefore, the objective was to identify the crucial residues involved, which could then be further investigated for protein bioengineering purposes to enhance its binding affinity for these heavy metals.
Regarding the comment “Moreover, authors listed microbial species used in bioremediation (line 98). Thus, the selection of the goal of the study and its connection with geographical location are not clear.”. Our selection of microbial species listed in the study, including Enterobacter cloacae, Pseudomonas fluorescens, Pseudomonas aeruginosa, Klebsiella oxytoca, Acinetobacter baumannii, Bacillus megaterium, and Corynebacterium glutamicum, was done to explore the orthologous relationships of genes, particularly metallothionein, across a range of bacterial species known for their roles in heavy metal bioremediation. This comparative genomics approach aimed to identify commonalities and distinctions in the genetic makeup of these species, providing insights into the potential of different bacteria for bioremediation purposes.
While our study didn't involve the direct isolation of P. aeruginosa strains from Madinah, it contributed valuable information on the genomic characteristics of this bacterium and other species in the context of heavy metal bioremediation. We acknowledge that the geographical location was not a primary focus of our study, but rather, our research was centered on the genetic and bioremediation aspects of these microorganisms. We hope this clarification helps to address the reviewer's query regarding the goals and geographical relevance of our study.
- Despite the certain results have been obtained in the article, their novelty is not obvious. Discuss several articles, which are focused on metal-binding proteins in P. aeruginosa, for example, 10.1107/S2059798321009608, https://doi.org/10.1107/S2059798321009608, https://doi.org/10.3390/ijms20205156, https://doi.org/10.1093/femsle/fnac071.
Compare the results obtained in the present work with those obtained in published works to reveal the novelty of your work.
Response 4: We appreciate the reviewer's thoughtful assessment and the opportunity to address the question regarding the novelty of our work in the context of other research articles.
Our research article focuses on investigating P. aeruginosa's potential for bioremediation of cadmium and lead pollution through the study of metallothionein protein interactions. We emphasize the critical role of metallothionein in binding these heavy metals and suggest the feasibility of protein engineering for enhanced heavy metal removal. Our study specifically targets cadmium and lead.
In response to the reviewer's valuable comment, we compared our work with the suggested research articles that delve into different aspects of P. aeruginosa's interactions with heavy metals, particularly zinc, and related compounds. Here's a summary of the comparisons:
The first referenced research article (DOI: 10.1107/S2059798321009608) explores zinc homeostasis in P. aeruginosa. It delves into the unique protein PA4063, which exhibits zinc-binding properties. PA4063's structural characteristics and zinc-binding affinity are highlighted, suggesting its potential roles as a zinc chaperone or concentration sensor. This work underscores the complexity of P. aeruginosa's responses to zinc deficiency.
The second referenced research article (DOI: https://doi.org/10.3390/ijms20205156) investigates the solute binding protein (SBP) repertoire of P. aeruginosa PAO1, consisting of 98 proteins. It employs bioinformatic predictions to propose overlapping ligand profiles for numerous SBPs, particularly for amino acids, polyamines, and quaternary amines. Experimental validation is conducted to identify specific ligands for certain SBPs, shedding light on the diverse ligand-binding capabilities of these proteins.
The third referenced research article (DOI: https://doi.org/10.1093/femsle/fnac071) examines Pseudomonas aeruginosa's resistance to calprotectin (CP), a protein that sequesters essential divalent metals, including zinc. The study reveals that strains lacking key zinc import systems can still thrive in the presence of CP. It introduces the concept of metallophores, such as nicotianamine, as facilitators of zinc acquisition by interfering with metal binding to proteins. This research extends our understanding of P. aeruginosa's adaptive responses in the context of innate immunity and metal sequestration.
Our main manuscript concentrates on the bioremediation of cadmium and lead, while the referenced research articles investigate various facets of P. aeruginosa's interactions with heavy metals, particularly zinc, and other compounds. Collectively, these studies contribute significantly to our comprehension of the bacterium's multifaceted biochemical responses and have implications for diverse research areas, including environmental remediation, chemotaxis studies, and innate immunity research. They collectively offer a more comprehensive view of P. aeruginosa's adaptive strategies in diverse contexts. We hope this clarifies the novel aspects of our work and its contributions to the existing body of research.
We have incorporated the following details and have cited the research articles in our discussion:
“Recent investigations have shed light on the intricate nature of P. aeruginosa's interactions with heavy metals and related compounds, emphasizing the significance of acknowledging this complexity. Although our main emphasis lies in exploring the bioremediation capabilities of this bacterium in relation to cadmium and lead, it is important to acknowledge the wider range of metal-related responses exhibited by this organism. Research on the protein PA4063 has revealed its distinctive zinc-binding characteristics, indicating potential functions as a periplasmic zinc chaperone or concentration sensor [58]. Additionally, the examination of solute-binding proteins (SBPs) in P. aeruginosa reveals their extensive range of abilities to bind different ligands, such as amino acids, polyamines, and quaternary amines [59]. Furthermore, the examination of the bacterium's resistance to calprotectin (CP) provides insights into its capacity to flourish in the existence of metal-sequestering proteins, potentially via the mechanism of metallophores [60]. The results of this study highlight the complex and flexible responses of P. aeruginosa to various metal challenges. These findings enhance our understanding of the interactions between this bacterium and metals, and their potential significance in bioremediation and environmental adaptation.”
Round 2
Reviewer 3 Report
In general, authors have improved the manuscript and add some novel information and references. Despite this, the main question remains unanswered: there are no direct connection between main environmental issue pointed in the text (heavy metal contamination in Madinah and selection of the goal of the study. Separately, these points are urgent and may be studied. Also, performed study presents valuable results, which possess scientific novelty.
Thus, improve the text. Add in the Introduction few paragraphs:
- on the importance of bioremediation technology for metal contaminated area in general and from the point of view of the situation in Madinah;
- with the situation on the use of microbial remediation technologies for metal contaminated areas, as well as on the importance of P. aeruginosa and related organisms for these methods.
Add additional references with this information.
Based on the information provided, improve the explanation of the goal of the study: point that P. aeruginosa may the organism, which may be potentially used for bioremediation in Madinah taking in account local environmental problems.
Author Response
Comment 1: In general, authors have improved the manuscript and add some novel information and references. Despite this, the main question remains unanswered:
there are no direct connection between main environmental issue pointed in the text (heavy metal contamination in Madinah and selection of the goal of the study.
Separately, these points are urgent and may be studied. Also, performed study presents valuable results, which possess scientific novelty.
Thus, improve the text. Add in the Introduction few paragraphs: - on the importance of bioremediation technology for metal contaminated area in general and from the point of view of the situation in Madinah; - with the situation on the use of microbial remediation technologies for metal contaminated areas, as well as on the importance of P. aeruginosa and related organisms for these methods. Add additional references with this information. Based on the information provided, improve the explanation of the goal of the study: point that P. aeruginosa may the organism, which may be potentially used for bioremediation in Madinah taking in account local environmental problems.
Response 1: We would like to express our gratitude to the reviewer for their constructive comments and suggestions. We have carefully considered their feedback and have made the necessary revisions to improve the manuscript.
To address the main concern regarding the connection between the environmental issue of heavy metal contamination in Madinah and the goal of our study, we have added additional paragraphs to the Introduction section. These paragraphs highlight the importance of bioremediation technology in general, particularly in metal-contaminated areas, and its relevance to the specific situation in Madinah. We have also included information on the use of microbial remediation technologies for metal-contaminated areas, emphasizing the significance of Pseudomonas aeruginosa and related organisms in such methods.
Furthermore, we have included additional references to support these discussions and provide a more comprehensive overview of the importance of bioremediation technology and the role of P. aeruginosa in metal-contaminated areas.
Based on the information provided, we have also improved the abstract and the explanation of the goal of our study. We now explicitly state that our study aims to investigate the potential use of P. aeruginosa for bioremediation in Madinah, considering the local environmental problems associated with heavy metal contamination. This clarification highlights the relevance and applicability of our research to the specific environmental challenges faced in Madinah.
We believe that these revisions have addressed the reviewer's concerns and have enhanced the clarity and coherence of the manuscript. We appreciate the reviewer's valuable input, which has undoubtedly improved the overall quality of our study. Thank you for your guidance and support throughout the review process.